# Neurodegenerative Changes in the Brains of the 5xFAD Alzheimer’s Disease Model Mice Investigated by High-Field and High-Resolution Magnetic Resonance Imaging and Multi-Nuclei Magnetic Resonance Spectroscopy

**DOI:** 10.3390/ijms24065073

**Published:** 2023-03-07

**Authors:** Chi-Hyeon Yoo, Jinho Kim, Hyeon-Man Baek, Keun-A Chang, Bo-Young Choe

**Affiliations:** 1Athinoula A. Martinos Center for Biomedical Imaging, Department of Radiology, Massachusetts General Hospital, Harvard Medical School, Charlestown, MA 02129, USA; 2Department of Health Sciences and Technology, Gachon Advanced Institute for Health Sciences & Technology, Gachon University, Incheon 21999, Republic of Korea; 3Neuroscience Research Institute, Gachon University, Incheon 21565, Republic of Korea; 4Department of Pharmacology, College of Medicine, Gachon University, Incheon 21936, Republic of Korea; 5Department of Biomedicine & Health Sciences, Research Institute of Biomedical Engineering, College of Medicine, The Catholic University of Korea, Seoul 06591, Republic of Korea

**Keywords:** Alzheimer’s disease, amyloid plaques, inflammation, high-resolution MRI, morphologic investigation, proton and phosphorus MRS, metabolic investigation

## Abstract

This study aimed to investigate morphological and metabolic changes in the brains of 5xFAD mice. Structural magnetic resonance imaging (MRI) and ^1^H magnetic resonance spectroscopy (MRS) were obtained in 10- and 14-month-old 5xFAD and wild-type (WT) mice, while ^31^P MRS scans were acquired in 11-month-old mice. Significantly reduced gray matter (GM) was identified by voxel-based morphometry (VBM) in the thalamus, hypothalamus, and periaqueductal gray areas of 5xFAD mice compared to WT mice. Significant reductions in N-acetyl aspartate and elevation of myo-Inositol were revealed by the quantification of MRS in the hippocampus of 5xFAD mice, compared to WT. A significant reduction in NeuN-positive cells and elevation of Iba1- and GFAP-positive cells supported this observation. The reduction in phosphomonoester and elevation of phosphodiester was observed in 11-month-old 5xFAD mice, which might imply a sign of disruption in the membrane synthesis. Commonly reported ^1^H MRS features were replicated in the hippocampus of 14-month-old 5xFAD mice, and a sign of disruption in the membrane synthesis and elevation of breakdown were revealed in the whole brain of 5xFAD mice by ^31^P MRS. GM volume reduction was identified in the thalamus, hypothalamus, and periaqueductal gray areas of 5xFAD mice.

## 1. Introduction

Alzheimer’s disease (AD), the most common neurodegenerative disorder, has been a leading cause of dementia and cognitive impairment in the elderly population [1]. Accumulations of extracellular amyloid-β (Aβ) and intracellular insoluble tau have been reported as hallmarks of AD [2]. Despite huge endeavors from researchers over decades, current treatments for AD remain palliative, while a definitive therapy to cure AD remains to be determined. Several transgenic rodent models have been used to develop and evaluate the potency of a new treatment for AD, which partially recapitulates the characteristics of AD [3,4]. A transgenic model of 5xFAD mice has been widely investigated, which exhibit aggressive amyloid deposition with gliosis beginning from two months and causing an enormous burden, especially in the subiculum and deep cortical layers [5]. Moreover, 5xFAD mice exhibit accumulation of intraneuronal Aβ_42_ at 1.5 months, amyloid plaques at two months, and neuronal loss at nine months [5]. Enormous efforts have been made using transgenic AD mice to identify molecular targets pivotal to the disease process and determine whether the modulation of the targets elicits a therapeutic effect [6,7]. However, frequent failure in translating preclinical studies to clinical trials indicates the importance of a deeper understanding of AD pathophysiology. Considering the growing body of neuroimage evidence in AD human brains, comparing in vivo brain profiles of transgenic AD mice to human AD patients might provide a deeper insight into the pathologic profile.

Metabolic alteration has been repeatedly reported in the brain of AD patients and transgenic animals. With different sets of detectable metabolites, proton (^1^H) and phosphorus (^31^P) resonance spectroscopy (MRS) have been used to measure regional changes in concentrations and ratios of metabolites associated with AD [8,9,10]. Typical ^1^H MRS findings are the significant reduction in N-acetyl aspartate (NAA) and elevation of myo-Inositol (mIns) in the hippocampus and cortical regions in the brain of the AD patients and animals models [8,9,11], postulated to decreased neuronal integrity and gliosis, respectively. Despite some conflicting results, preclinical studies have investigated changes in glutamate (Glu), γ-aminobutyric acid (GABA), glutathione (GSH), and taurine (Tau) in transgenic models, reporting alterations in these metabolites [12]. Previous ^31^P MRS studies have revealed altered concentrations of high-energy phosphate, adenosine triphosphate (ATP), and energy buffer phosphocreatine (PCr) in the brain of AD patients [13,14]. Moreover, significant changes in the ratio of membrane phospholipids, phosphomonoesters (PME), and phosphodiesters (PDE) in the AD brain have been documented over progression of the disease, indicating AD-related disturbance in membrane turnover [15].

Efforts have been made to investigate in vivo evidence of the structural changes of well-reproduced features in post mortem AD brain tissues. Magnetic resonance imaging (MRI) has been widely used to detect morphologic changes in the AD brain. Remarkably, voxel-based morphometry (VBM) was proposed to provide an effective and robust method of comparing brain structures between different populations [16,17]. After its initial success, VBM has been increasingly used to investigate AD-induced morphologic changes in the brain in cross-sectional and longitudinal ways [18]. It has been repeatedly reported that middle temporal lobes, specifically the entorhinal cortex, are involved in AD pathology from the early stage, spreading out to other gray matter (GM) regions, including the hippocampus, parietal, and frontal areas, with disease progression [18,19,20,21]. In addition, ventricular enlargement and decreased brain volumes are reported in the AD brain [18,19,20,21]. Despite the success of VBM in clinical studies, applications of VBM to transgenic animals are still restricted. Due to differences between human and rodent brains, direct application of the existing VBM methods, optimized for human brains, is challenging [22,23,24]. Moreover, the small size of mice brains raises another technical challenge in acquiring high-resolution MRI.

Previously, several studies investigated disease-related changes in the brain of 5xFAD transgenic mice using MRI, MRS, and immunofluorescence. Increased mIns and decreased NAA levels were reported in 5xFAD mice by Mlynarik et al. measured by ^1^H and ^31^P MRS. However, an integrated study of VBM and multi-nuclei (^1^H-^31^P) MRS has yet to be performed for 5xFAD mice. This study aimed to investigate morphological and metabolic changes in the brain of 5xFAD mice using high-resolution MRI, in vivo ^1^H, ^31^P MRS, and immunofluorescence. The pathologic profile completed by integrating automated volumetric analysis and quantification of the metabolite concentrations will provide a deeper understanding of disease pathways in 5xFAD mice and their link to the human AD brain.

## 2. Results

### 2.1. VBM Analysis

A representative MRI (10-month-old WT mouse) is illustrated in Figure 1 (top row), with the tissue templates of GM, WM, and CSF (from the second row to bottom). The MRI scans with unacceptable quality (i.e., ghosting and aliasing) were excluded, and the VBM analysis was performed for twelve 10-month-old mice (5xFAD, n = 6; WT, n = 6) and fifteen 14-month-old mice (5xFAD, n = 6; WT, n = 9). The co-registration of the mice in each age group was reliably performed, and each mouse’s resulting modulated GM volumes and TBVs were used for the VBM analysis. The region-based comparison of the TBV and modulated GM volumes are listed in Table 1, showing no significant differences between WT and 5xFAD mice. The voxel-wise comparison is illustrated in Figure 2. A significant decrease in GM volumes was identified in the hypothalamic and thalamic regions of 10-month-old 5xFAD mice compared to WT, illustrated with the population-based template of mice brains (Figure 2a). Larger clusters of GM reduction were identified in the hypothalamus and thalamus of 14-month-old 5xFAD mice and periaqueductal gray areas compared to WT (Figure 2b).

### 2.2. ^1^H MRS

Figure 3 illustrates representative ^1^H MRS scans of (a, b, e, f) 5xFAD and (c, d, g, h) WT mice at (a, b, c, d) 10 months of age and at (e, f, g, h) 14 months of age. The MRS scans with unacceptable quality and high CRLB value (> 20%) were excluded. Table 2 lists the concentrations of the brain metabolites obtained in the PFC of the 10-month-old mice (5xFAD, n = 6; WT, n = 7) and 14-month-old mice (5xFAD, n = 7; WT, n = 8), respectively. Compared to WT, 10-month-old 5xFAD mice showed statistically significant (*p* < 0.050) reduction in GSH, Tau, tNAA, and tCr. However, no significant difference was observed in 14-month-old 5xFAD mice. The concentrations of the brain metabolites measured in the hippocampus of the 10-month-old mice (5xFAD, n = 7; WT, n = 8) and 14-month-old mice (5xFAD, n = 6; WT, n = 9) are listed in Table 2. Trend-level reductions in GABA (*p* = 0.066) and tNAA (*p* = 0.071) and an increase in mIns (*p* = 0.063) were observed in 10-month-old 5xFAD mice compared to WT. At the age of 14 months, 5xFAD mice showed a significant reduction in tNAA (*p* = 0.029) and an increase in mIns (*p* = 0.001), and a trend-level decrease in GABA (*p* = 0.098) and an increase in Tau (*p* = 0.063) compared to WT.

### 2.3. ^31^P MRS

Figure 4 illustrates ^31^P MRS scans obtained in the whole brain region of the 11-month-old (a) 5xFAD and (b) WT mice. The PCr peak was at the center of the frequency range, and all peaks of the individual metabolites can be identified. Table 2 lists the/PCr of the metabolites from the 5xFAD (n = 7) and WT mice (n = 5). Statistically significant decreases in NADP and PME/PDE ratio were observed in the 5xFAD mice compared to WT (*p* < 0.050).

### 2.4. Immunofluorescence

To confirm the changes in neuronal and glial cells, we performed the immunofluorescence staining in the mice brains at 10 and 14 months of age, respectively. The brain tissues were stained with NeuN and GFAP and Iba1 with Thioflavin S for double labeling staining and, particularly, analyzed for the dentate gyrus of the hippocampus and cortex. For 10 months of age, a significant decrease in the NeuN intensity was observed in the cortex (*p* = 0.0176) and dentate gyrus (*p* = 0.0003) of the 5xFAD mice compared to the WT mice (Figure 5a,b). In addition, the GFAP intensity was significantly increased in the cortex (*p* = 0.0396) of the 5xFAD mice compared with the WT mice (Figure 5a,c). However, no significant difference was observed in the GFAP intensity of the dentate gyrus between the 5xFAD and WT mice (Figure 5a,c). For the Iba1 intensity of the 5xFAD mice, significant increases were observed in the cortex and dental gyrus (*p* = 0.0045; *p* = 0.0102) compared with the WT mice (Figure 5d,e). The number of amyloid plaques assessed by Thioflavin S stain was observed in the cortex (*p* = 0.0001) and dentate gyrus (*p* < 0.0001) of the 5xFAD mice but not WT mice (Figure 5d,f). For 14 months of age, the neuronal loss (*p* = 0.0221) and astrogliosis (*p* = 0.0046), represented as NeuN and GFAP intensity changes, were observed in the cortex of 5xFAD mice compared with WT mice (Figure 5g–i). In addition, NeuN intensity (*p* = 0.0028) was significantly decreased in the dentate gyrus of the 5xFAD mice (Figure 5g,h). Interestingly, the dentate gyrus of 5xFAD mice also showed an elevated intensity of GFAP (*p* = 0.0208, Figure 5g,i). In the Iba1 and Thioflavin S staining, the significantly elevated intensities of microglia (*p* = 0.0352, Figure 5j,k) and the amyloid plaques (*p* = 0.0016, Figure 5j,l) were observed in the dentate gyrus of 5xFAD mice. Our results found that the neuronal loss and astrogliosis, initially observed in the 5xFAD mice brains at 10 months of age, were exacerbated concerning aging.

## 3. Discussion

This study aimed to investigate metabolic and morphologic changes in the brain of 5xFAD mice at the same time. The VBM analysis was applied to determine whether a volumetric loss, a typical finding in AD patients, is replicated in the brain of 10- and 14-month-old 5xFAD mice. The PFC and hippocampus metabolites were quantified by ^1^H MRS and compared between 5xFAD and WT mice. In addition, the level of ATP, PCr, and membrane phospholipids were measured by in vivo ^31^P MRS to reveal whether significant changes in the high energy phosphates and membrane phospholipids occur in the 5xFAD mice brain. Immunofluorescence was performed to validate that 5xFAD mice develop the known neuropathology, such as neuronal loss, amyloid plaque deposition, and plaque-associated gliosis.

Growing numbers of studies have used automated VBM analysis to detect volumetric changes in the AD brains [18]. Previous studies revealed that the structural abnormalities consistently encompassed the entorhinal cortex and hippocampal regions, which can be predictive as patients progress from mild cognitive impairment to AD [25]. On the other hand, only a limited number of studies investigated the morphologic changes in the brains of 5xFAD mice, and the results still need to be more robust. Large differences in size, shape, and image contrasts between human and rodent brains have hindered automated VBM analysis implementation for rodent brains. This study applied several pre-processing steps with SPM8 automated VBM analysis to investigate the morphometric changes in in vivo brain of 5xFAD mice [23]. The population-based template was reliably generated for each age group, as previously described [23]. The regional-based comparison revealed no significant differences between 5xFAD and WT mice. The voxel-wise analysis identified clusters around the thalamus and hypothalamus where the GM volume of the 10-month-old 5xFAD mice was reduced compared to WT. Larger clusters were detected in the thalamus, hypothalamus, and periaqueductal gray areas of the 14-month-old 5xFAD mice, compared to WT. Previous studies with 5xFAD mice at the 2, 4, and 6 months reported no observable brain volume loss in the forebrain, cerebral cortex, ventricles, frontal cortex, hippocampus, striatum, or olfactory bulbs [26,27]. Although methodological differences exist between studies, i.e., automated VBM vs. manual delineation, our results may indicate that the volume loss in 5xFAD mice occurs in relatively late steps (~10 months). Macdonald et al. reported significant hippocampal volume reduction in 5xFAD mice at 13 months of age and significantly reduced [^18^F]Fludeoxyglucose uptakes throughout the brain compared to age-matched WT mice [28], while our results showed no observable volume loss in the hippocampus of the 5xFAD mice. Differences between methods, such as magnetic field strength (9.4T vs. 3T), resolution (100 μm^3^ vs. 142 μm^3^), and analysis, hinder a meaningful comparison of the results. Our VBM results in the 5xFAD mice, significant GM reductions in the thalamus, hypothalamus, and periaqueductal gray did not agree with the findings in human AD brains, which remains to be determined.

Several preprocessing was applied to improve the spectral quality in this study. Obtained ^1^H MRS scans were separately stored (16 × 20) and averaged (320) after the spectral registration, incorporating frequency and phase correction [29], shown in Figure 3. The CRLB values of the metabolites in Table 2 addressed that the quantification was reliably performed for the metabolite of interest in the hippocampus and PFC. A trend level reduction of NAA was observed in the hippocampus of the 10-month-old 5xFAD mice, which reached a statistical significance when the 14-month-old 5xFAD were compared to WT. A significant reduction of NAA was observed in the PFC of the 10-month-old 5xFAD mice compared to WT, which is consistent with the findings in the AD human brain [8,9], and in the brain of Tg2576 [12,30] and 5xFAD [31,32] transgenic model. The immunofluorescence demonstrated that NeuN-positive cells in the cortex and hippocampus were significantly reduced in 10- and 14-month-old 5xFAD mice compared to WT. Thus, the depletion of NAA and neuronal losses, consistently reported neurometabolic changes, was replicated in this study. Nearly significant increases of mIns were found in the hippocampal of the 10-month-old 5xFAD mice, which reached a statistical significance at 14 months of age, compared to WT. Elevated mIns is often interpreted with increased glial cells and consistently reported in the hippocampus of AD patients [8,9]. In addition, previous studies with the 5xFAD mice reported increased mIns in the hippocampus of the 5xFAD mice compared to WT [31,32]. Consistently, we observed the elevation of mIns and gliosis in the hippocampus of 5xFAD mice, supported by the immunofluorescence results, in which Iba1 and GFAP-positive cells in the cortex and hippocampus were significantly increased in the 10- and 14-month-old 5xFAD mice compared to WT. Trend level decreases of GABA were observed in the hippocampus of the 10- and 14-month-old 5xFAD mice compared to WT. Previously, a reduction in GABA+ were observed in AD patients’ frontal and parietal cortex compared to age- and gender-matched controls, respectively [33]. In addition, a significant reduction of GABA was observed in the hippocampus of 5xFAD mice compared to WT [31]. Given that GABA is primarily located in neurons, a decreasing trend of GABA in 5xFAD mice may imply loss or dysfunction of GABAergic neurons. Interestingly, a reduction of GSH was observed in the PFC of the 10-month-old 5xFAD mice, while that of the 14-month-old 5xFAD mice was elevated compared to WT. The alteration of GSH levels have been reported in AD brains [34,35,36], suggesting that GSH plays a vital role in the pathological pathway of AD [37]. Although the elevation of GSH in 5xFAD mice remains undetermined, reduction of GSH was consistent with the previous study, which reported a significant decrease of GSH/Cr in the cortical region of the Tg2576 mice compared to the control [12]. Depletion of GSH might lead neurons to be vulnerable to oxidative stress-induced damage. Changes of Tau in 5xFAD mice were different between the regions, such as a significant elevation in the hippocampus at 14 months and decrease in the PFC at 10 months. Although the basis for this observation remains to be determined, one possibility is that osmoregulatory homeostasis was disrupted similar to the previous studies [38,39,40].

Localized in vivo ^31^P MRS scans were acquired in the whole brain of mice, as shown in Figure 4. Although some previous ^31^P MRS studies have revealed significant changes in high-energy phosphate, such as, ATP, PCr, or Pi level in the various regions of the AD brain [13,14], conflicting results have been reported [41]. In this study, no significant changes were observed in these metabolites, ATP/PCr or Pi/PCr, consistent with the previous studies [31,41]. Similar to the previous findings, this result may reveal that a significant depletion in brain energy stores, chronic hypoxia, nor ischemia has occurred in the brain of 5xFAD mice [31]. Decreased PME with elevated PDE in 5xFAD mice was consistent with the previous study, which reported a significant decrease of PME and an increase of PDE in bilateral hippocampi of mild cognitive impairment and AD subjects compared to control subjects [42]. Moreover, previous studies reported that the alteration in brain PME and PDE levels varies according to the progress of AD [14,15]. In the early stage of the AD brain, PME levels were reported to be increased [15], while in the progressed AD brain, PDE was elevated in the brain [14]. The 5xFAD mice in this study is known to resemble the progressed AD. Thus, the alteration of PME and PDE levels in the 5xFAD mice brain is consistent, at least partially, with this previous clinical findings [14]. Considering their roles, decreased PME and elevated PDE indicate a disruption in the membrane synthesis and elevation of breakdown in the brain of 5xFAD mice [14,42]. Significant reduction in NADP in the brain of 5xFAD mice can be interpreted that the redox status being increased, considering that the most critical function of NADP is to counteract oxidative damage and to maintain a pool of reducing equivalent essential for other detoxification reactions [43].

There is a limitation in this study. First, rather than the VBM-identified clusters where 5xFAD mice showed reduced GM volume compared to WT, anatomical regions (hippocampus and PFC) were used to measure the metabolite concentrations and for group comparisons. Since MRI and ^1^H MRS scans need to be acquired within the same scan, the clusters were not identified at the moment of acquiring spectra for each animal. Although limited spatial coverage of ^1^H MRS remains a limitation of this integrated experimental protocol, the VBM results indicate that the metabolic alteration in 5xFAD mice is not driven by global brain volume changes. Second, not all scans were acquired with a follow-up scheme in the same animals. Natural loss of animals occurred for both groups during the interval between 10 months and 14 months of their age. Age-matched animals in the same batch were carefully added to the 14-month-old dataset to compensate for the number loss and maintain statistical power for interpretation. Considering a prolonged scan time burden on the animal, MRI/^1^H MRS scans and ^31^P MRS could not be acquired simultaneously. A future study that reaches all scans in the same animal longitudinally is expected to integrate morphologic and metabolic investigation with minimum inter-subject variability. Lastly, the relative metabolic ratio (/PCr) was used for ^31^P MRS. Referring to the previous findings, we used PCr as an internal reference assuming the PCr levels to be stable throughout the whole brain region of 5xFAD mice and WT. In a future study, the absolute concentration of PCr can be measured by ^1^H MRS with enhanced spectral resolution. It can calibrate the concentrations of ^31^P MRS metabolites, which will be another advantage of this integrated experimental protocol.

## 4. Materials and Methods

### 4.1. Animal Preparation

The Institutional Animal Care and the Use Committee of the Lee Gil Ya Cancer and Diabetes Institutional Center, which abides by the Institute of Laboratory Animal Resources, approved all animal experiments (LCDI-2019-0155). The 5xFAD transgenic mice, which express five human APP (Swedish mutation: K670N, M671L; Florida mutation: I716V; London mutation: V717I) and PS1 genes (M146L, L286V)], were used at 10/11/14 months of age with age-matched wild-type (WT) mice. Food and water were provided *ad libitum*, and the breeding room’s photoperiod, temperature, and humidity were automatically maintained at 12L:12D, 21 ± 2 °C, and 50 ± 10%.

### 4.2. Acquisition of MRI/MRS

All MRI and ^1^H/^31^P MRS were obtained using a 9.4T BioSpec^®^ system (Bruker BioSpin Corporation, Billerica, MA, USA), operated with ParaVision^®^ 6.0 (Bruker BioSpin Corporation, Billerica, MA, USA), as illustrated in the Appendix A. The system was equipped with a 210-mm horizontal bore main magnet, and 400-mT/m actively shielded gradients with integrated shim coils. A linearly polarized transmit resonator (112/72 mm) and a 4-channel phased-array surface coil were used to obtain MRI and ^1^H MRS scan in the 10- and 14-month-old mice brains. The animals were anesthetized using the spontaneous inhalation of 2.0–2.5% isoflurane and a 1:2 mixture of O2:NO2 (250 mL/min). The animals were positioned in the prone position in the handling system. A pressure sensor (Bruker BioSpin Corporation, Billerica, MA, USA) was used for respiratory monitoring, and the body temperature of the animals was stabilized at 37.5 °C using a waterbed heater system (Bruker BioSpin Corporation, Billerica, MA, USA). High-resolution T2-weighted images were acquired using TurboRARE sequence with the following parameters: repetition time (TR) = 2000 ms; effective-echo-time (TE_eff_) = 33 ms; rapid acquisition relaxation enhanced (RARE) factor = 16; averages = 2; field of view = 12 × 12 × 15.6 mm^3^; matrix size = 120 × 120 × 156; resolution = isotropic 100 μm. Two volumes of interest (VOIs) of ^1^H MRS contained the prefrontal cortex (PFC; 2.0 × 1.5 × 1.2 mm^3^) and unilateral hippocampus (left; 2.0 × 1.2 × 2.0 mm^3^). An automated shimming was performed to achieve an unsuppressed water linewidth of 11–15 Hz. A point-resolved spectroscopy (PRESS) sequence with a sinc3-90° excitation pulse (12,420 Hz; 0.5 ms) and two of sinc3-180° refocusing pulses (4650 Hz; 1.0 ms) were used with the following parameters: TR/TE = 4000/15.02 ms; complex data points = 2048; average = 16 × 20 (20 sets of 16-average spectra); spectral bandwidth = 5000 Hz. Localized in vivo ^31^P MRS scan was acquired in the whole mouse brain (7.0 × 4.0 × 7.0 mm^3^) at 11 months of age, using a ^1^H-^31^P dual-tuned coil. Automatic shimming was used to achieve an unsuppressed water linewidth of 40–46 Hz. Full 3D image selected in vivo spectroscopy (ISIS) sequence with a bp32 excitation pulse (16,000 Hz; 0.08 ms) and calculated-180° inversion pulses were used with the followings: TR = 4000 ms; complex data points = 4096; average = 128; spectral bandwidth = 16,025.64 Hz.

### 4.3. VBM Analysis

The acquired MRI was processed using the VBM8 toolbox implemented in the Statistical Parametric Map (Wellcome Trust Centre for Neuroimaging, UCL Institute of Neurology, London, UK; www.fil.ion.ucl.ac.uk/spm (accessed on 24 July 2022)) software with multiple processing steps dedicated to the mice brain [23,24]. The population-based templates were separately generated for 10-month-old (n = 12) and 14-month-old mice (n = 15), as previously described [23]: (1) The voxel size of the image was multiplied by 10. (2) All animals’ images were linearly registered to the strain-specific mouse template [23] using Advanced Normalization Tools (ANTs). (3) The skull was stripped using the brain mask of the strain-specific mouse template. (4) Intensity bias correction was applied using the ANTs’ N4biasFieldCorrection [44] within the brain mask. (5) The bias-corrected brain image was again registered to the mouse brain template. (6) The brain image was segmented into GM, white matter (WM), and cerebrospinal fluid (CSF) using a mixture of Gaussians and tissue probability maps (TPMs). Total brain volume (TBV) was calculated by summing the number of voxels in each animal’s GM and WM classes. (7) Each mouse’s GM and WM images were used to generate a population-based GM and WM template. (8) The modulated GM volume image was generated from the deformation field obtained from the non-linear registration from each animal to the population template. (9) The outcomes were compared between the WT and 5xFAD groups for each age using the general linear model (GLM)/univariate analysis (SPM8) with one-tailed *t*-statistics inferring voxels of lower intensity in 5xFAD mice compared to WT using the TBV as the covariance of no interest. Similar to the previous studies which used rodent VBM analysis [45,46,47], the statistical significance was considered *p* < 0.01 (uncorrected for multiple comparisons) with a minimum cluster size of 500 voxels. Cluster locations were identified with an anatomical atlas of the mouse brain [48].

### 4.4. Preprocessing and Quantification for MRS

Preprocessing steps were performed to improve the spectral quality using the FID Appliance (FID-A) open-source software package (Version 1.2; https://github.com/CIC-methods/FID-A (accessed on 28 May 2022)). A total of 20 sets of the raw spectra were apodized with a 2-Hz exponential filter, and a time-domain spectral registration was applied. The aligned spectra were averaged into a single spectrum, corresponding to 320 averages. Quantifying ^1^H MRS was conducted using the linear combination of model (LCModel; Version 6.3-0I; http://s-provencher.com/lcmodel.shtml (accessed on 28 May 2022)). The free induction decay (FID) signals of the preprocessed ^1^H MRS scans were analyzed in 0.20 to 4.30 ppm range by fitting the spectra with the individual peaks of the metabolites from the parametrically matched simulated basis sets. To simulate the basis set, the FID-A software was used with the PRESS sequence, and the following parameters: resonance frequency (400.31 MHz) and TE values (TE1/TE2, 7.926/7.090) were selected to simulate the spectra in the range of 12.53–−3.13 ppm (spectral bandwidth of 5000 Hz) with 32,768 data points and a linewidth of 2-Hz. The following metabolites were included in the simulated basis set: Alanine, aspartate, creatine (Cr), GABA, Glu, glutamine (Gln), GSH, glycerophosphocholine (GPC), lactate, mIns, NAA, N-acetylaspartylglutamate (NAAG), PCr, phosphorylcholine (PCh), scylloinositol, and Tau. The chemical shifts and J-coupling constants were obtained from Govindaraju et al. [49]. For the absolute quantification of ^1^H MRS, the unsuppressed water signals were used as an internal reference with a brain water concentration of 80%. In addition, the receiver gain was scaled by inputting the factor of [Gain_unsuppressed_/Gain_suppressed_] [50]. The metabolite’s Cramér-Rao lower bound (CRLB) was calculated as a reference for the quantification reliability.

Preprocessing and quantification of ^31^P MRS were performed using java-based magnetic resonance user interface (jMRUI; Version 5.2; http://www.mrui.uab.es/mrui (accessed on 28 May 2022)) software. The acquired FID signals were apodized with a 6-Hz Lorentzian filter, and an automated zero and first-order phase correction was applied. The resonance frequency of PCr was assigned to the central frequency of 0.0 ppm. A non-linear least-squares quantitation algorithm, the AMARES algorithm, was used for the quantification [51]. Resonances of PCr, γ-ATP, α-ATP, β-ATP, nicotinamide adenine dinucleotide phosphate (NADP), GPC, glycerophosphoethanolamine (GPE), inorganic phosphate (Pi), phosphoethanolamine (PE), and PCh, were visually identified in the full spectra in conjugate with the prior knowledge of the line parameters from the previous studies. The resonances were assigned as individual peaks and quantified with the automated zero and first-order phase correction for 4096 data points. The fitted amplitude of all metabolites was scaled by that of PCr. The CRLB value of the metabolites was calculated by dividing the standard deviation by its signal amplitude. In addition, the PCr relative amplitude (/PCr) of total ATP (tATP; γ-ATP + α-ATP + β-ATP), PME (PCh + PE) and PDE (GPC + GPE), and PME/PDE ratio were calculated.

### 4.5. Immunofluorescence and Thioflavins S Staining

After the scan, 10- or 14-months aged mice were anesthetized with a mixture of Zoletil (16.7 mg/kg) and Rompun (15.5 mg/kg). The brain was perfused with normal saline containing heparin, fixed using 10% formalin overnight, and then dehydrated in 30% sucrose solution at 4 °C for three days. The brain tissues were frozen and cut into 22 μm-thick slices using a cryostat (Cryotome, Thermo Electron Corporation, Waltham, MA, USA). The slices were stored in the cryoprotectant solution at 4 °C until the staining. Immunostaining and Thioflavin S staining were performed on floating sections. After being washed in PBS-T (0.2% Triton X-100), the brain slices were blocked in PBS-T containing 0.5% BSA and 3% normal horse serum. They then were double-stained with the NeuN (Millipore, Burlington, MA, USA, 1:500) and GFAP (Dako, Glostrup, Hovedstaden, Denmark, 1:500) or with Iba1(Novus, Englewood, Colorado, 1:500) and Thioflavin S. The brain slices were washed in PBS-T and incubated with Goat-anti-rabbit IgG Alexa Fluor 488 (Invitrogen, Waltham, CA, USA), Goat anti-Mouse IgG Alexa Fluor 555 (Invitrogen), or Goat anti-Rabbit IgG Alexa Fluor 555 (Invitrogen) antibody. For Thioflavin S staining, sections were stained for 10 min with a 0.0125% Thioflavin S solution in 30% ethanol. The slices were washed and mounted onto slides using Antifade Mounting Medium with DAPI (Vector Laboratories, Burlingame, CA, USA). Specimens were examined using the TS2-S-SM microscope (Nikon Microscopy, Tokyo, Japan). Serial images of 40× and 100× magnification were captured on four sections per animal. Each group was compared and analyzed by region of interest (ROI) intensity ratio (%) and the number of amyloid plaques stained by Thioflavin S using NIS-Elements software (4.40.00 64-bit, Nikon). Once the ROIs were defined, the green channel showing Alexa Fluro 488 and the red channel showing Alexa Fluro 555 was used to measure the intensity of the green or red signal within each ROI per section (n = 3–5 per group).

### 4.6. Statistical Analysis

IBM SPSS Statistics 21 (IBM Corporation, Armonk, NY, USA) software was used for all statistical analyses. The TBV and regional modulated GM volumes were compared between the 10- and 14-month-old WT and 5xFACE mice. The metabolite concentrations in the hippocampus and PFC, measured by ^1^H MRS, were compared between the WT and 5xFAD mice, using independent-sample *t*-tests, respectively, for the 10 and 14 months of age. Moreover, the/PCr of the metabolites were compared between the 5xFAD and WT mice using independent-sample *t*-tests. In the results of immunofluorescence, statistical analysis was performed using GraphPad Prism 9.1.0 (221) software (GraphPad Software Inc., San Diego, CA, USA), and outliers were removed using the Outlier calculator (significance level: Alpha = 0.05) in GraphPad Prism software. Differences in the collected data between groups were analyzed using the unpaired *t*-test. The data are presented as the mean ± standard error of the mean (SEM). All regional statistical analyses considered differences with *p* values less than 0.05 statistically significant.

## 5. Conclusions

Both 10- and 14-month-old 5xFAD mice have been known to have an onset of disease before the beginning of the experiment. Commonly reported ^1^H MRS features, reduced NAA, and elevated mIns were replicated in the hippocampus of the 14-month-old 5xFAD mice compared to WT, supported by the immunofluorescence results. Decreased PME and elevated PDE were observed in the whole brain of 5xFAD mice compared to WT, which can be interpreted as a sign of disruption in the membrane synthesis and elevation of breakdown. Surprisingly, a significant GM volume reduction was identified in the thalamus, hypothalamus, and periaqueductal gray areas of 5xFAD mice, which remains to be determined.

## Figures and Tables

**Figure 1 ijms-24-05073-f001:**
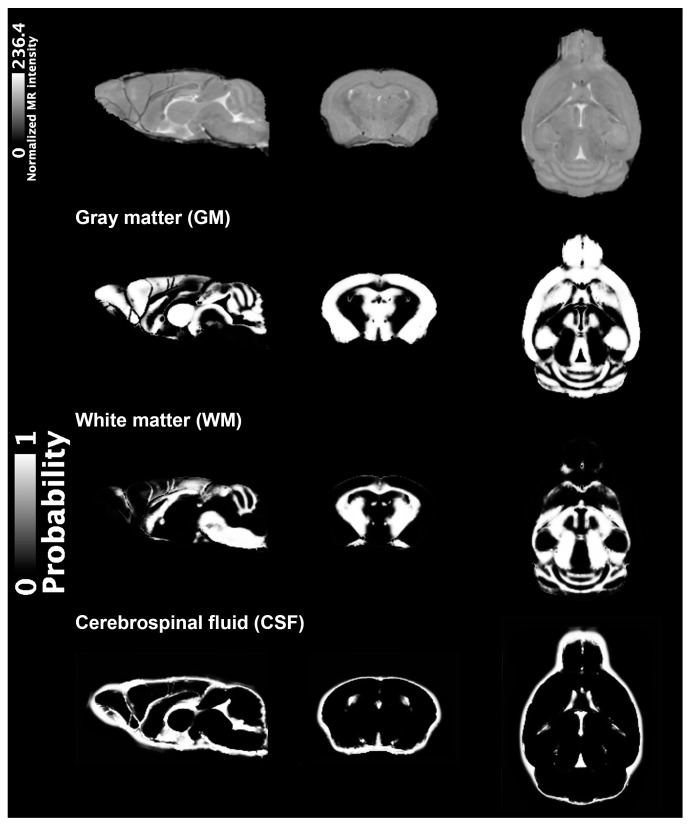
Representative 3D T2-weight MR images of a WT mouse at 10 months of age (top row) acquired using TurboRARE sequence with the following parameters (skull stripped): TR/TEeff = 2000/33 ms; RARE factor = 16; averages = 2; field of view = 12 × 12 × 15.6 mm^3^; matrix size = 120 × 120 × 156; resolution = 100 μm isotropic, illustrated with the generated population-based tissue probability maps visualizing GM, WM, and CSF separately. Magnetic resonance, MR; wild-type, WT; repetition time, TR; effective echo time, TEeff; rapid imaging with refocused echoes, RARE; gray matter, GM; white matter, WM; cerebrospinal fluids, CSF.

**Figure 2 ijms-24-05073-f002:**
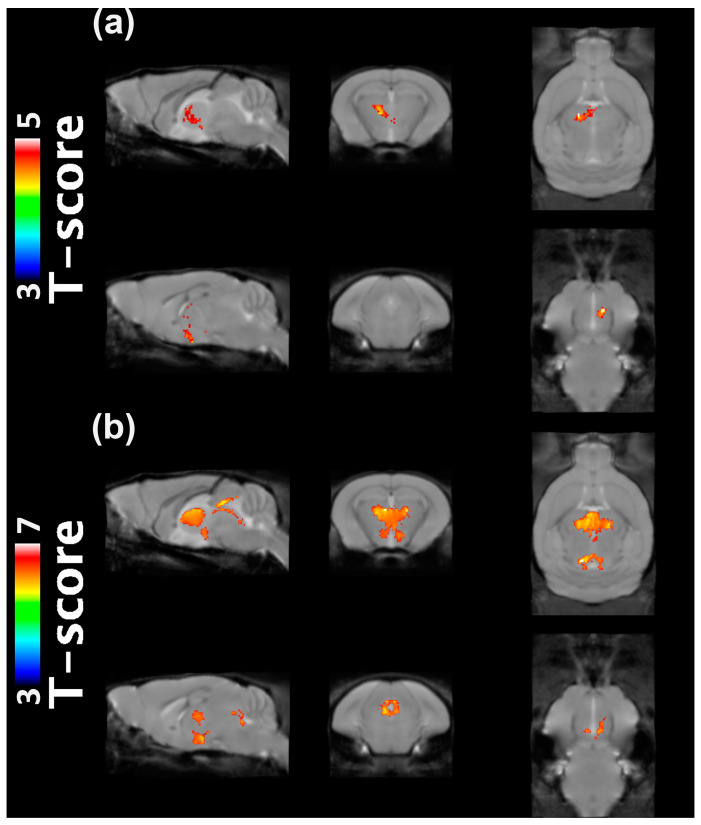
Automated VBM analysis results (SPM8) comparing the modulated GM volumes between WT and 5xFAD mice at (**a**) 10 months and (**b**) 14 months of age. The clusters (red) where GM volume was significantly decreased in 5xFAD mice compared to WT mice are illustrated with the population-based template of mice brains for anatomical reference. Statistical significance was determined as *p* < 0.01 with a minimum cluster size of 500 voxels. Voxel-based morphometry, VBM; gray matter, GM; wild-type, WT.

**Figure 3 ijms-24-05073-f003:**
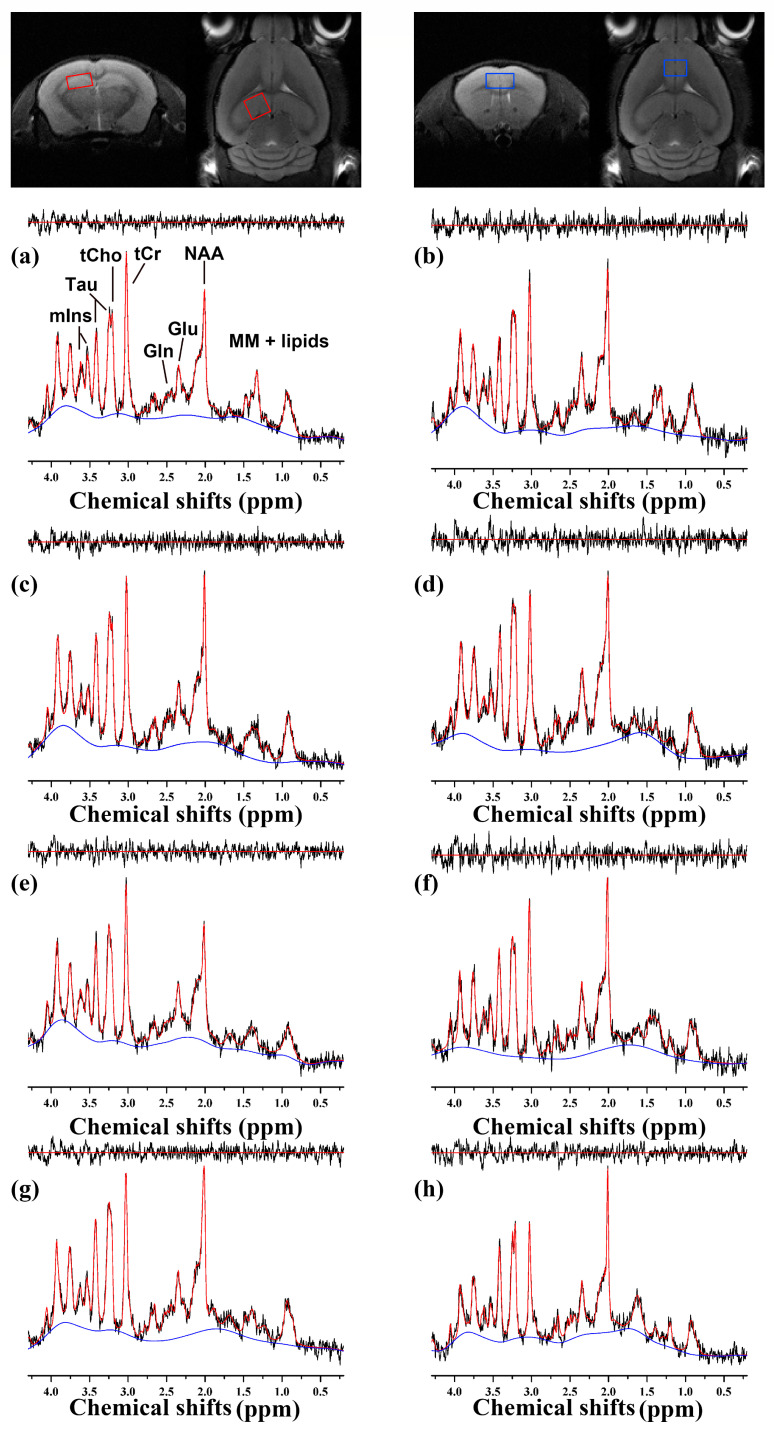
Representative ^1^H MRS scans, obtained using PRESS sequence with the following parameters: TR/TE—4000/15.016 ms; complex data points—2048; average—16 × 20; spectral bandwidth:—5000 Hz. Scans in the left hippocampus (2.0 × 1.2 × 2.0 mm^3^; red box) of the (**a**) 5xFAD and (**c**) WT mice at 10 months of age and (**e**) 5xFAD and (**g**) WT mice at 14 months of age and in the PFC (2.0 × 1.5 × 1.2 mm^3^; blue box) of the (**b**) 5xFAD and (**d**) WT mice at 10 months of age and (**f**) 5xFAD and (**h**) WT mice at 14 months of age were illustrated as the raw spectra (black), fitted spectra (red), spectra baseline (blue), and residual error (top). Proton magnetic resonance spectroscopy, ^1^H MRS; point-resolved spectroscopy, PRESS; repetition time, TR; echo time, TE; prefrontal cortex, PFC.

**Figure 4 ijms-24-05073-f004:**
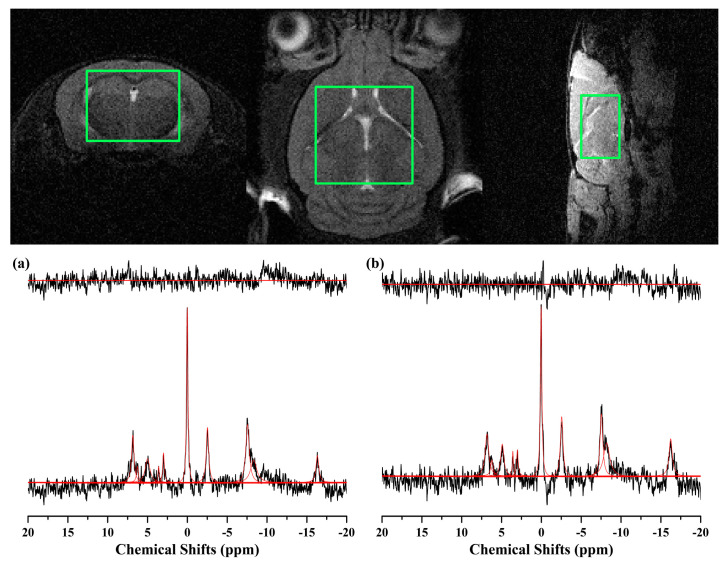
Representative ^31^P MRS scans, obtained using ISIS sequence with the following parameters: TR = 4000 ms; complex data points = 4096; average = 128 (1024); spectral bandwidth = 16,025.64 Hz. Scans in the whole brain of mice (7.0 × 4.0 × 7.0 mm^3^; green box) of the (**a**) 5xFAD and (**b**) WT mice at 11 months of age were illustrated as the raw spectra (black), fitted spectra (red), spectra baseline (blue), and residual error (top). Phosphorus magnetic resonance spectroscopy, ^31^P MRS; image selected in vivo spectroscopy, ISIS; repetition time, TR.

**Figure 5 ijms-24-05073-f005:**
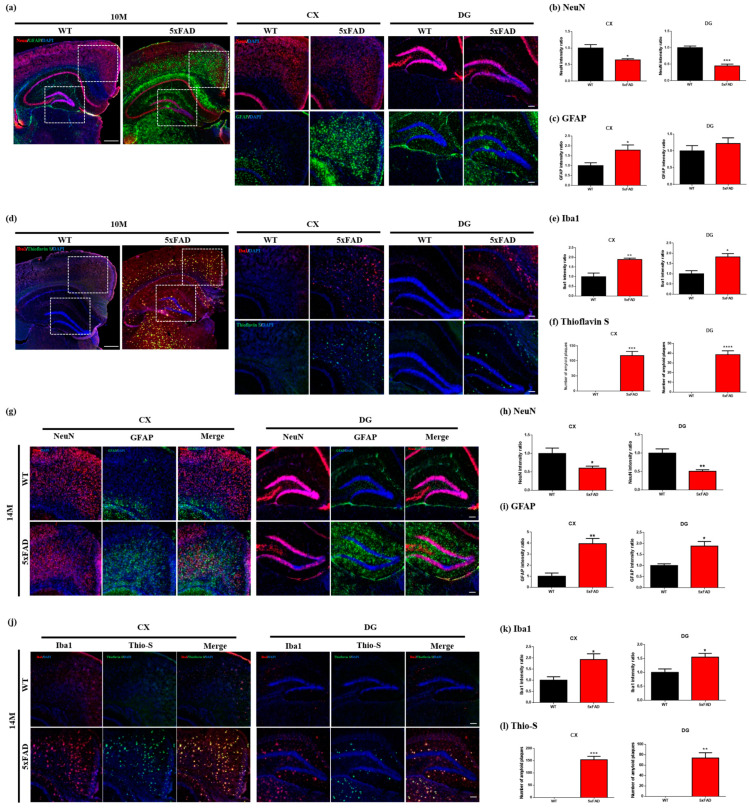
(**a**) Double immunostaining of WT and 5xFAD mice brains (10 months of age (n = 4)) with NeuN- and GFAP-specific antibodies. The cortex and dentate gyrus were observed in the white rectangle. (**b**) The intensities of (**b**) NeuN and (**c**) GFAP were quantified in the cortex and dentate gyrus of mice. (**d**) Iba1 targeted to the microglia was stained with Thioflavin S in WT and 5xFAD mice brains (10 months of age (n = 4)). (**e**) In the cortex and dentate gyrus of mice, the intensity of (**e**) Iba1 and (**f**) Thiflavin S was quantified. (**g**) Double immunostaining of WT and 5xFAD mice brains (14 months of age (n = 3–5)) with NeuN- and GFAP-specific antibodies. (**h**,**i**) Quantification of intensity was measured in the cortex and dentate gyrus. (**j**) In the brain of 14-month-old mice (n = 3–5), microglia with amyloid plaque was stained with Iba1 and Thioflavin S. The intensity of (**k**) Iba1 and (**l**) Thioflavin S was quantified in the cortex and dentate gyrus. Magnification 40×; scale bars, 500 μm. Magnification 100×; scale bars, 100 μm. Mean ± SEM. * *p* < 0.05, ** *p* < 0.01, *** *p* < 0.001, **** *p* < 0.0001 vs. CTL.

**Table 1 ijms-24-05073-t001:** Global brain volume estimates and regional modulated GM volumes of the 10- and 14-month-old WT and 5xFAD mice.

	10-Month-Old				14-Month-Old			
	WT		5xFAD		WT		5xFAD	
Global Measures	Average	SD	Average	SD	Average	SD	Average	SD
GMV	0.24	0.01	0.25	0.02	0.24	0.01	0.22	0.08
ICV	0.56	0.08	0.53	0.03	0.57	0.05	0.49	0.17
TBV	0.38	0.01	0.39	0.02	0.39	0.01	0.35	0.13
Brain regions	Modulated GM volume							
Caudate-putamen	67.62	22.9	73.11	24.81	67.83	24.02	68.15	24.19
Globus pallidus	16.35	14.4	18.5	16.21	15.69	14.38	15.18	13.88
Hippocampus	62.78	20.9	69.68	22.87	65.97	21.47	64.12	20.92
Amygdala	68.76	21.18	75.55	22.93	71.74	21.83	69.79	21.39
Thalamus	31.95	29.6	33.45	30.81	32.18	29.71	32.2	29.77
Hypothalamus	48.48	26.12	49.34	26.71	48.73	25.84	48.59	25.8
Neo cortex	59.69	23.49	65.9	25.45	61.25	24.09	60.87	24.02
Central gray	55.98	17.54	56.86	18.54	56.28	16.89	54.23	16.73
Cerebellum	38.87	20.54	41.62	21.9	39.04	20.92	38.37	20.36
Midbrain	15.48	14.24	15.56	14.75	15.36	14.47	14.99	14.15
Olfactory bulb	60.45	26.4	59.17	26.61	60.09	25.48	60.36	25.4
Internal capsule	12.7	9.89	14.65	11.22	12.14	10.13	11.78	9.53
External capsule	47.49	21.7	52.88	23.75	49.47	22.25	48.9	22.06
Anterior commissure	67.22	12.48	70.58	12.51	66.7	12.37	68.24	12.07
Superior colliculi	19.17	13.87	20.46	15.1	20.1	14.43	19.1	13.88
Inferior colliculi	15.47	13.59	16.94	15.13	17.29	14.74	16.86	14.37
Fimbria	15.2	7.08	16.56	7.45	14.55	6.19	14.57	6.09
Basal forebrain septa	57.91	18.44	62.33	19.47	59.82	18.18	60.03	18.57

Abbreviations: GM = gray matter; WT = wild type; SD = standard deviation; GMV = gray matter volume; ICV = intracranial volume; TBV = total brain volume.

**Table 2 ijms-24-05073-t002:** Absolute and relative concentrations of the brain metabolites obtained from in vivo ^1^H and ^31^P MRS.

	In Vivo ^1^H Spectroscopy							
	Cortex				Hippocampus			
	10 Month		14 Month		10 Month		14 Month	
Metabolites	5xFAD	WT	5xFAD	WT	5xFAD	WT	5xFAD	WT
GABA	3.67 ± 0.67	3.49 ± 0.70	2.90 ± 0.71	3.18 ± 0.72	2.80 ± 0.88	3.74 ± 0.93	2.79 ± 1.24	3.86 ± 0.75
Gln	4.90 ± 1.15	5.39 ± 1.17	5.06 ± 1.06	5.13 ± 0.98	4.37 ± 0.99	4.06 ± 0.67	3.99 ± 1.01	4.29 ± 0.56
Glu	13.07 ± 1.27	13.61 ± 1.45	13.56 ± 1.15	12.68 ± 1.18	8.38 ± 0.81	8.74 ± 0.93	8.92 ± 1.12	9.08 ± 0.55
GSH	1.77 ± 0.31	2.52 ± 0.49	2.44 ± 0.69	1.81 ± 0.42	1.74 ± 0.55	1.93 ± 0.36	2.34 ± 0.48	1.99 ± 0.40
mIns	6.93 ± 0.45	7.01 ± 0.85	7.01 ± 1.43	6.20 ± 0.89	6.92 ± 0.86	6.16 ± 0.41	7.92 ± 0.83	6.18 ± 0.67
Tau	13.82 ± 1.41	15.80 ± 1.08	15.07 ± 2.50	15.00 ± 1.29	11.83 ± 1.71	11.75 ± 0.95	13.71 ± 1.02	12.32 ± 1.45
tCho	2.24 ± 0.56	2.46 ± 0.22	2.12 ± 0.57	2.26 ± 0.25	1.75 ± 0.34	1.71 ± 0.13	1.65 ± 0.37	1.88 ± 0.17
tNAA	10.37 ± 0.55	11.53 ± 1.05	10.39 ± 0.78	11.07 ± 1.07	8.27 ± 0.64	8.94 ± 0.68	8.57 ± 0.92	9.60 ± 0.71
tCr	9.49 ± 0.88	10.80 ± 1.05	10.44 ± 2.25	10.06 ± 1.07	10.03 ± 0.70	9.36 ± 0.71	10.46 ± 1.01	10.09 ± 0.96
Glx	17.96 ± 1.57	18.99 ± 2.02	18.62 ± 1.74	17.81 ± 1.79	12.75 ± 1.53	12.81 ± 1.25	12.90 ± 1.76	13.37 ± 0.93
	In vivo ^31^P spectroscopy in the whole brain							
	11 month							
Metabolites	5xFAD	WT						
tATP/PCr	1.86 ± 0.22	1.78 ± 0.15						
Pi/PCr	0.40 ± 0.13	0.45 ± 0.08						
PME/PCr	0.66 ± 0.07	0.76 ± 0.03						
PDE/PCr	0.36 ± 0.07	0.26 ± 0.06						
NADP/PCr	0.54 ± 0.14	0.81 ± 0.11						

Abbreviations: MRS = magnetic resonance spectroscopy; WT = wild-type; GABA = γ-aminobutyric acid; Gln = glutamine; Glu = glutamate; GSH = glutathione; mIns = myo-Inositol; Tau = taurine; tCho = total choline; tNAA = total N-acetylaspartate; tCr = total creatine; Glx = glutamate complex; PCr = phosphocreatine; CRLB = Cramér–Rao lower bound; ATP = adenosine triphosphate; NADP = nicotinamide adenine dinucleotide phosphate; PDE = phosphodiesters; PME = phosphomonoesters; Pi = inorganic phosphate.

## Data Availability

All data and materials are present in the paper and available upon request.

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
