# Peer review of "Neurodegenerative Changes in the Brains of the 5xFAD Alzheimer’s Disease Model Mice Investigated by High-Field and High-Resolution Magnetic Resonance Imaging and Multi-Nuclei Magnetic Resonance Spectroscopy"

_ijms, 2023, doi:10.3390/ijms24065073_

Round 1
Reviewer 1 Report
The article by Yoo et al., concerns the purpose of setting up and applying MR imaging protocols for combined spectroscopic analysis of a commonly used mouse Alzheimer´s model to obtain human disease relevant insight to the disease.
The study intent is highly relevant and the experimental imaging protocols are well described with high quality data shown.
Also, the manuscript is well written and structured making it easy to read and build on experimentally.
I am critical to the sparce information about the model and the recruited animals. The lack of method information makes it difficult to judge the quantitative conclusions made in the manuscript and hence the biological findings that the authors go to great lengths to discuss relative to literature on the human disease.
This leaves me with the impression that the authors have produced a solid and well performed study relevant to build new studies on where especially the possibility to combine two different individually strong MRSI methods with the VBM method is exiting. BUT, without an experimental scheme to understand what is possible and what is not possible to combine in one examination and without careful information about which animals and how they have been treated makes it difficult to judge the biological/biochemical value of the study. This latter criticism also leads to wonder about the value of the lengthy literature-based discussion.
Specifically
In materials and methods:
Please provide details on the mice included in the study: strain identification, weight, sex; especially the weight (brain volume) could have direct influence on the amounts of quantified metabolites and thus directly on the conclusions drawn
Which anesthetics and concentrations of these have been used?
How were the animals monitored and what was the temperature variation group-wise?
What was the magnetic field strength of the Bruker system?
An overview (Figure) of the experimental scheme (i.e. what was done to the individual animals and when was it done). This is needed to understand the potential of investigating different parameters (31P, 1H MRSI and 1H MRI) in the same animal and in general to evaluate the findings.
Discussion:
The discussion is lengthy and provides a nice literature overview, but it is not easy to extract the new findings provided in the manuscript. I suggest shortening and make sure to highlight the main findings and divide the mouse references from the references to human studies.
Minor language issues:
Line 255 remove the word not
Lines 269-270 the reference to detoxification reactions is not adequate
Reviewer 2 Report
This study by Chi-Hyeon Yoo, Jinho Kim, and Colleagues presented morphological and metabolic assessment of the 5xFAD mice brains. Though the technical details may sound difficult for the general audience, I must appreciate the author's efforts in constructing the manuscript (MS) in a read-through style. This MS would be an excellent reference for future studies implying MRI and MRS in AZ models. Each section of the MS is well-detailed, nicely written, and easy to follow. I enjoy reading.
I have two minor concerns;
1) Reg. Figure 2
Top right of a and b-do they belong to one mouse at a different age? similar to 5xFAD panel.
Even though they are different mice, the top right 'a' and the top right 'b' has many identical spots, and even the background shows a lot of similarities. Same with the 5xFAD panel. I request the authors to clarify.
2) Reg. Figure 5
Its difficult to understand the figure. It would be helpful if authors draw a rectangle (for CX, DG) also in WT. DAPI (nuclei) staining is poor, especially d-WT. DAPI signal is almost absent in d-WT, however, it is prominent in enlarged WT-DG.
Could you provide more information on how the images are acquired and analyzed? Specifications of the light source (epifluorescence or confocal laser)? Did the authors use any isotype control staining to normalize the fluorescence signal? how the background/autofluorescence is subtracted?
